# Visual Field Restriction in the Recognition of Basic Facial Expressions: A Combined Eye Tracking and Gaze Contingency Study

**DOI:** 10.3390/bs14050355

**Published:** 2024-04-23

**Authors:** Melina Boratto Urtado, Rafael Delalibera Rodrigues, Sergio Sheiji Fukusima

**Affiliations:** 1Faculty of Philosophy, Sciences and Letters at Ribeirão Preto, University of São Paulo, Ribeirão Preto 14040-901, Brazil; fukusima@usp.br; 2Institute of Mathematics and Computer Science, University of São Paulo, São Carlos 13566-590, Brazil; rafaeldr@usp.br

**Keywords:** visual inspection, facial expression recognition, eye tracking, gaze contingency, moving window technique

## Abstract

Uncertainties and discrepant results in identifying crucial areas for emotional facial expression recognition may stem from the eye tracking data analysis methods used. Many studies employ parameters of analysis that predominantly prioritize the examination of the foveal vision angle, ignoring the potential influences of simultaneous parafoveal and peripheral information. To explore the possible underlying causes of these discrepancies, we investigated the role of the visual field aperture in emotional facial expression recognition with 163 volunteers randomly assigned to three groups: no visual restriction (NVR), parafoveal and foveal vision (PFFV), and foveal vision (FV). Employing eye tracking and gaze contingency, we collected visual inspection and judgment data over 30 frontal face images, equally distributed among five emotions. Raw eye tracking data underwent Eye Movements Metrics and Visualizations (EyeMMV) processing. Accordingly, the visual inspection time, number of fixations, and fixation duration increased with the visual field restriction. Nevertheless, the accuracy showed significant differences among the NVR/FV and PFFV/FV groups, despite there being no difference in NVR/PFFV. The findings underscore the impact of specific visual field areas on facial expression recognition, highlighting the importance of parafoveal vision. The results suggest that eye tracking data analysis methods should incorporate projection angles extending to at least the parafoveal level.

## 1. Introduction

Due to its biological, psychological, and social significance, the processing of facial expressions has been the subject of behavioral, neurophysiological, and even computational experiments. Building upon the seminal work of Darwin [1] and motivated by studies conducted by the physiologist Sir Charles Bell, as well as extensive research by Ekman and colleagues [2,3] and Carroll [4], it is now established that humans can express and recognize at least six different emotional states, happiness, sadness, anger, fear, surprise, and disgust, with remarkable cross-cultural stability.

Facial expressions are characterized by the activation of muscle groups surrounding internal facial regions such as the eyes, nose, and lips [1,5,6]. Consequently, the eye and lip regions have been considered highly relevant for emotional expression recognition. Research conducted using eye tracking techniques has found that most fixations are concentrated within these facial regions, and the visual inspection movement patterns vary according to different emotional facial expressions. The eye region provides more information for the identification and recognition of anger and sadness, whereas happiness and disgust expressions are more recognizable when inspected in the lip region. Fear and surprise expressions depend on both regions [7,8,9]. These findings suggest that directing foveal vision and visual attention to specific regions, rather than others, may be crucial for emotion recognition and discrimination. However, in other studies, the proportion of fixations on diagnostic facial areas was less consistent or unaffected [10,11]. These inconsistencies render the evidence regarding selective visual inspection and specific facial characteristics inconclusive.

Understanding the possible existence of a visual inspection pattern and selective attention to emotional faces in healthy groups would be a significant advancement in this field, as dysfunctional visual inspection has been indicative of developmental psychopathologies, such as autism spectrum disorder [12], affective psychopathologies like schizophrenia [13], social phobia [14], and neurological impairments [15]. This understanding could guide clinical research or even aid in the development of diagnostic methods for these populations.

Despite the advancements facilitated by eye tracking technology in enhancing and standardizing the research methodologies, discrepancies in the results persist, even among studies utilizing this equipment. Many current studies employ eye tracking data analysis methods that predominantly prioritize the examination of foveal vision [16]. Notably, a common oversight in most of these studies is the failure to account for potential overlaps in visual fields near the boundaries of the areas of interest (AOIs), compounded by the lack of standardization in AOI definition and replication. This scenario arises when a fixation is interpreted as being inside an AOI, but the relevant information is actually captured from a neighboring AOI due to the extension of the visual field for the same fixation. This issue has been highlighted by Orquin et al. [17], who indicated that the absence of a proper margin between AOIs could potentially compromise conclusions regarding attentional cognitive processes, although not specifically within the context of emotional face recognition. In their work, the authors provide guidelines for the use of AOIs in eye tracking research, including the recommendation for AOI margins of $1^{{}^{\circ}}$ to $1.5^{{}^{\circ}}$ of the visual angle, aligning with the size of the fovea. However, despite their valuable contribution in enhancing the quality of eye tracking experimental design and data analysis, their recommendation does not consider the potential influences of simultaneous parafoveal and peripheral information.

Given the properties of foveal and parafoveal vision, which are fundamental for our understanding of visual processing, it is important to consider their roles in cognitive tasks. While foveal vision covers about $1^{{}^{\circ}}$ to $2^{{}^{\circ}}$ of our visual field and provides sharp detail and color perception when we focus directly on an object, parafoveal vision extends to about $5^{{}^{\circ}}$ and is better suited to detecting motion and shapes, albeit with less detail [18]. Therefore, investigating the role of visual field areas to assess their impact on emotional facial expression recognition is of significant importance.

In light of this consideration, employing eye tracking combined with the gaze contingency technique allows for the artificial constraining of visual information, enabling focused study in these areas. This approach could shed light on improved directives for eye tracking methods of data analysis, including AOI definition and other important parameters, which are primarily reliant on the foveal visual field. Ultimately, it could aid in clarifying the inconclusive evidence presented in the literature. Thus, our work aims to combine eye tracking with the gaze contingency procedure to artificially restrict participants’ visual fields. Accordingly, we seek to investigate the impacts on visual inspection measures and the judgment of emotional facial expressions for five emotions (depicting happiness, sadness, anger, fear, and neutrality) under three distinct conditions: no visual restriction (NVR), parafoveal and foveal vision (PFFV), and foveal vision (FV).

Eye tracking has been extensively utilized in the realm of neuroscience, proving particularly effective in investigating perceptual and facial recognition processes. The methodology encompasses a suite of technologies dedicated to extracting information about eye movements, fixations, and saccades. Specifically, the eye tracking apparatus employs corneal reflection technology, capturing key indicators such as pupil positioning, dilation, and constriction [16]. By emitting infrared rays, the equipment captures the corneal reflection through micro-cameras, facilitating the extraction of eye movement data during unconstrained visual exploration.

The gaze contingency paradigm, originally developed by Reder [19] to investigate reading comprehension, provides insights into the input processes of eye movements involved in scene and facial inspection when combined with eye tracking. More specifically applicable to our research context, the moving window technique further refines eye tracking techniques by allowing the separation of what is fixated from what is processed, as it restricts the viewer’s visual angle, giving the sensation of looking through a telescope. Therefore, the gaze contingency technique serves as a method to control the information fed into the visual system and isolate information outside the visual angle, allowing for the computational isolation of areas of the visual field.

In our literature search, we found limited studies in which the moving window technique (or gaze contingency) was used to investigate emotional face perception. For example, Atkinson and Smithson [20] examined the recognition of facial expressions with diagnostic areas specifically allocated to the foveal region. However, the presentation of the expressions was brief, consisting of only one fixation, and was specifically mapped to the foveal region. Kim et al. [21] explored how the search for specific diagnostic regions contributed to the recognition of facial emotions, utilizing the moving window technique to prevent peripheral visual processing. However, a possible limitation was the absence of real eye movement tracking, as the moving window was controlled with a computer mouse, rather than by tracking the participants’ gaze. Additionally, it is important to note that this study included only female participants, potentially introducing bias, as demonstrated in prior research [22,23]. In a study by Birmingham et al. [24], the general aim of the research was to investigate differences in eye movement patterns between young and elderly populations. The experimental design included only expressions of anger, fear, and happiness, and it did not aim to investigate the visual inspection of all four basic facial expressions [25], as we do in this study. Lastly, Danion and Flanagan [26] employed both the moving window technique and eye tracking, although their research did not focus on emotional face perception but rather on distinguishing gaze strategies to track moving targets with the eyes and hands.

The rationale for combining eye tracking and the moving window technique lies in the recognition that eye tracking alone may not definitively reveal which information is genuinely relevant in facial processing, as there are three distinct types of visual processing at play: foveal, parafoveal, and peripheral. Consequently, by restricting the visual field, we can delve into how participants process emotional faces. If the visual inspection of facial expressions relies on specific facial features (foveal vision), as some studies have shown, we hypothesize that restricting the visual field will not significantly impact this process and might even enhance the observation. If this hypothesis holds, we expect that the restriction of the visual field may make the recognition of facial expressions more demanding in time but will not compromise the accuracy. Furthermore, we explore the inspection time when viewing basic facial expressions among groups exposed to both restricted and unrestricted conditions. This exploration is grounded in previous studies that have demonstrated differences in the recognition time of basic facial expressions. It is worth noting that the combination of eye tracking and the gaze contingency experimental procedure represents a novel approach to studying the recognition of the four basic facial expressions, as these methods have been more commonly applied to investigate facial identity [27,28,29], objects [30], and scenes [31,32].

## 2. Materials and Methods

### 2.1. Participants

A total of 163 volunteers, comprising both genders (81 females and 82 males), with an average age of 23.2 (SD 4.7), participated in the research. These 163 volunteers were randomly assigned to three groups with distinct experimental conditions (NVR = 54, PFFV = 59, FV = 50). The number of participants was justified based on an a priori power calculation conducted using the G*Power 3.1.9.7 [33], which indicated that a total of 159 participants was sufficient to detect a medium effect (Cohen’s d=0.25) at α=0.05, with power of 0.80. Following an invitation to participate in the research, the participants were adequately informed about the nature of the study and read and signed an informed consent form approved by the Research Ethics Committee (CEP) of the University of São Paulo.

All participants had normal or corrected vision, assessed through a visual acuity test and self-report questionnaires. The exclusion criteria encompassed (a) participants with vision problems that could interfere with the study’s outcome, (b) participants with low visual acuity (verified through the Freiburg Visual Acuity Test), (c) participants who demonstrated poor performance in the eye tracking equipment calibration phase (with calibration errors exceeding $1^{{}^{\circ}}$, corresponding to performance of less than three stars), and (d) participants exhibiting impairments related to neurological or psychiatric disorders. In total, data from 12 participants were excluded from the study, none of which were included in the previously reported statistics on the 163 participants. One was due to the loss of the data-recording files, one due to insufficient fixations, and ten due to outlier eye movement measures. The excluded participants were already deducted from the previously given values.

The participants were recruited through verbal invitations extended by the researchers within the university environment, as well as through announcements in university newspapers, social media, and flyers posted on university premises. The demographic data indicated that 76% of the participants were undergraduate students, 20% were graduate students, and 4% were professionals with some affiliation with the university or were friends of students. It is pertinent to emphasize that participation was entirely voluntary, and no participant received any form of financial compensation for their involvement in the study.

### 2.2. Materials and Equipment

Eye movements were recorded using The Eye Tribe’s, Copenhagen, Denmark, desktop binocular eye tracking device, model ET1000, configured to operate at its maximum data acquisition rate (60 Hz). Visual stimuli were displayed on a LG, Seoul, South Korea, Flatron 23MP65HA monitor (1920 × 1080 pixel resolution, 23 inch screen) positioned 57 cm from the observer’s face. The experiment was conducted through PsychoPy-2022.2.3, a Python-based interface that provides precise control over visual stimuli and improved data accuracy [34].

Visual acuity was assessed using the Freiburg Visual Acuity and Contrast Test (FrACT10) software version 1.0, displayed on an LG Flatron E2241PX monitor (1920 × 1080 pixel resolution, 21.5 inch screen). The “C” Landolt test was employed to measure visual acuity [35].

Thirty emotional facial images were presented as visual stimuli, equally distributed among the five emotional classes: happiness, sadness, anger, fear, and neutrality. We justified the selection of these emotions based on the research findings of Jack et al. [25], who identified evidence for four well-discriminated basic expressions. Their study, utilizing a method combining perceptual expectation modeling, information theory, and Bayesian classifiers, suggested that fear and surprise share processing and representation codes, as do disgust and anger, particularly at the early processing stages, aligning with the reported patterns of major recognition confusion in categorization tasks [36]. Furthermore, these four emotions are encompassed within Ekman’s six classical emotions [3,37]. In our study, the alternative theory of four basic expressions reduces confusion in emotion recognition by prioritizing more identifiable emotions [25,36,38], which is crucial in enhancing the effects stemming from changes in visual field restriction for the recognition of facial expressions. Finally, the inclusion of a neutral face is common practice in facial expression recognition studies, serving as a neutral and stable reference for comparison with emotional expressions [36].

All images were of frontal faces and were also equally distributed among the genders (male and female), resulting in 15 images for each gender, with 3 facial images per gender–emotion. These images were sourced from the Karolinska Directed Emotional Faces (KDEF) image database [39,40]. The images were originally 562 × 762 pixels but were edited to grayscale and displayed at 843 × 1143 pixels, which approximated the size of a real face [41]. The selected facial images were oriented frontally, improving the diagnostic region inspection. The individuals in the image database were Caucasian and approximately 30 years old, similar to the participants in this study.

### 2.3. Procedure

Primarily, this research was conducted under three independent experimental conditions: no visual restriction (NVR), parafoveal and foveal vision (PFFV), and foveal vision (FV). The distinction between these conditions lay in the level of visual field restriction imposed on the participants. In the NVR condition, stimuli were presented without any restriction, while, in the PFFV condition, the visual field was limited to the parafovea and fovea, covering $5^{{}^{\circ}}$ of the central visual angle. In the FV condition, the visual field was restricted solely to the fovea, covering only $2^{{}^{\circ}}$ of the visual angle.

The visual restriction was implemented using the moving window technique, directly controlled by the eye tracker in response to the viewer’s eye gaze. A mask was applied over the stimulus, with a visual aperture diameter corresponding to the projection of the respective visual angle on the screen, $5^{{}^{\circ}}$ for PFFV or $2^{{}^{\circ}}$ for FV. The diameter of the aperture for each condition was calculated beforehand based on the viewer’s distance and the monitor specifications (screen size and resolution), implying that the PFFV condition exhibited a larger aperture hole compared to the FV condition. It is worth noting that each participant was randomly assigned to only one of the three experimental conditions, resulting in three independent experimental groups.

The experiments were conducted in a well-lit and noise-free room to minimize any disturbances or alterations to these conditions. After reading and signing the informed consent form, the participants were instructed to complete a questionnaire containing demographic questions related to their background and health. The participants also completed the Toronto Alexithymia Scale (TAS-26), a validated Brazilian version designed for university populations [42]. After these procedures, each participant was led to a chair positioned one meter away from the computer screen to perform a computerized visual acuity test [35], entering their responses using an adapted numeric keyboard.

Following the visual acuity test, each participant was directed to another chair, where they were asked to sit and carefully listen to the experiment’s instructions. Only after the participant had a complete understanding of the instructions did the researcher position them in the chair with their chin resting at a distance of 57 cm from the monitor. The face-to-monitor distance was defined by the manufacturer of the eye tracker equipment as a basic protocol to ensure data recording quality. The chin rest was used to minimize head movements, which could interfere with the eye tracking data collection process by altering the coordinate system established during calibration. Additionally, the participants were instructed to place their arms on the table to provide better support and reduce movement during the experiment.

The experimental design of the collection procedure is illustrated in Figure 1. It began with the calibration of the eye tracker, aimed at establishing an appropriate coordinate system for accurate eye tracking. This process considered the individual characteristics of each participant, such as the facial geometry, ocular anatomy, interocular distance, and eye positioning on the face. After satisfactory calibration, which involved displaying 16 points at different positions on the computer screen for approximately 20 s, the recording of the coordinates (x,y) representing the participant’s gaze projection on the screen commenced. After the calibration process, a screen displayed the experiment’s instructions, directing the participant to press a key when ready to begin.

The subsequent screen presented a central fixation point, displayed for one second, intended to ensure the participant’s initial focus alignment before each stimulus’ presentation. Following this, the emotional face image stimulus was presented. In the PFFV or FV condition, the face image could only be inspected through a small moving window aperture, with its diameter defined according to the experimental condition ($5^{{}^{\circ}}$ for PFFV or $2^{{}^{\circ}}$ for FV); refer to the fourth screen from the top in Figure 1a. For the NVR condition, the stimulus screen showed the full facial image with no restriction; refer to the fourth screen in Figure 1b.

Each face remained on the screen for as long as the participant decided, advancing to the next stage with a press of the SPACE key on the keyboard. The choice of a free presentation time was justified by the scarcity of studies comparing emotional face judgment under different visual field restriction conditions. Although previous research has shown that facial recognition can be achieved with few fixations [43], our approach enabled the acquisition of a comparative dimension regarding the impact of participant judgment, facilitating a more comprehensive and meaningful analysis of the participants’ responses across the various experimental contexts.

After the presentation of the facial stimulus, the participant answered a single question: “What did the face seem to express?” (refer to the fifth screen from the top in Figure 1a,b). Then, one of the perceived expressions was selected using the numeric keyboard, among the options “Happiness” (1), “Sadness” (2), “Neutrality” (3), “Fear” (4), and “Anger” (5). Subsequently, the participant was asked to indicate their degree of confidence in their response, as “Yes, sure” (1), “Yes, with doubt” (2), “I can’t say” (3), “No, with doubt” (4), or “No, sure” (5), also using the numeric keyboard; refer to the last screen in Figure 1a,b.

The set of screens, comprising the central fixation point, facial stimulus, and subsequent judgment, was repeated 30 times, with each repetition constituting a single trial in the participant’s session. The facial images were randomly presented without repetition, equally distributed among five emotional expressions (happiness, sadness, anger, fear, and neutrality), resulting in six images per emotional class. This number of images was aimed to mitigate visual fatigue due to the longer duration of the experiment [44]. It should be noted that the set of images remained consistent across the experimental groups (NVR, PFFV, FV). It is also important to emphasize that, during the visual inspection in the PFFV and FV conditions, the eye tracking induced visual field restrictions (via the moving window paradigm), controlling the information input while simultaneously recording the eye movements. In contrast, the NVR condition only recorded the eye movements. Finally, the experiment, in its entirety, lasted approximately 60 min for each participant.

### 2.4. Data Analysis and Measures

The statistical analysis encompassed various metrics, including the number of fixations, fixation duration, visual inspection time, and recognition accuracy, treated as dependent variables, while NVR, PFFV, and FV were considered independent variables. Facial expressions were categorized as factors influencing these dependent variables, allowing for a statistical data analysis segmented by facial expression. Due to the non-normal distribution of the data and unsuccessful transformations owing to the lack of positive skewness, we employed the Kruskal–Wallis test, a non-parametric method, throughout the data analysis. To assess the observed differences among groups, we calculated the effect sizes using the eta-squared ($\eta^{2}$). The resulting values of $\eta^{2}$ were 0.01, 0.06, and 0.14, indicating small, moderate, and large influences of the groups on the variability of the dependent variables, respectively [45,46]. Subsequently, Dunn’s test was applied as a post hoc analysis. A significance level of *p* = 0.05 was employed, with Bonferroni correction assuming an adjusted *p* = 0.01.

#### 2.4.1. Accuracy

Accuracy was calculated by comparing each participant’s response to the expected emotional expression for each presented face, divided by the total number of stimuli (trials) or by the number of images in each expression category, as applicable. This assessment gauged the participant’s effectiveness in task performance.

#### 2.4.2. Consensus

The assessment of confidence aimed to examine the level of certainty that participants placed in their choices of emotions for each category of facial expressions. To achieve this, confidence levels were expressed on a Likert scale consisting of five points: “Yes, sure” (1), “Yes, with doubt” (2), “I can’t say” (3), “No, with doubt” (4), “No, sure” (5). For the specificity of the metric analysis and standardization, we adopted the following terminology: “Strongly agree” (1), “Agree” (2), “Not sure” (3), “Disagree” (4), “Strongly disagree” (5). These categories were assigned the following ordinal values: SA = 1, A = 0.5, N = 0, D = −0.5, SD = −1. It is important to note that this representation did not alter the distribution of the data. The Likert scale was converted into
(1)$$X=\left\{\begin{array}{c} -1,-0.5,0,0.5,1 \end{array}\right\}$$
where the numerical values constitute a symmetrical scale, ensuring equal intervals between options. Based on the information obtained from the Likert scale, three metrics were considered for analysis: mean, consensus (Cns), and entropy (Ent).

The mean was calculated with the following equation [47]:(2)$$\mu_{x}=\sum_{i=1}^{n}p_{i}X_{i}$$
where $p_{i}$ represents the relative frequency (or, in appropriate contexts, the probability) of Likert scale options, and *n* is the total number of response categories in the Likert scale.

The semantic counterparts resulting from the means were characterized by the following intervals:(3)$$\begin{array}{cccc} \{1.00\geq & \mu_{x}\;>0.75\} & \; & =\;SA\\ \{0.75\geq & \mu_{x}\;>0.25\} & \; & =\;A\\ \{0.25\geq & \mu_{x}\;\geq -0.25\} & \; & =\;N\\ \{-0.25> & \mu_{x}\;\geq -0.75\} & \; & =\;D\\ \{-0.75> & \mu_{x}\;\geq -1.00\} & \; & =\;SD \end{array}$$

The entropy was calculated with the following equation [47]:(4)$$Ent\left(X\right)=-\sum_{i=1}^{n}p_{i}log_{2}\left(p_{i}\right)$$

The Shannon entropy value indicates the variability in the data, capturing the level of uncertainty or information content present within the dataset, as opposed to the traditional measure of variability, such as the standard deviation in normal distributions, which primarily reflects the dispersion of the data around the mean.

The consensus was calculated with the following equation [47]:(5)$$Cns\left(X\right)=1+\sum_{i=1}^{n}p_{i}log_{2}\left(1-\frac{|X_{i}-\mu_{x}|}{d_{x}}\right)$$
where $\mu_{x}$ and $dx=X_{max}-X_{min}$ represent the mean and width of *X*, respectively.

The Likert scale data were treated as an ordinal categorical variable, using the Cns measure proposed by Tastle and Wierman [47]. According to the authors, Cns can be understood as complete agreement reached by a group of individuals, also known as general agreement. Thus, the closer the Cns value was to 1, the greater the consensus among the participants regarding the certainty of their choice of emotional expression for each face (complete agreement). Conversely, the closer the value was to −1, the less consensus there was regarding the certainty of their choice of emotional expression for the face (complete disagreement). In the Cns measure, a shift in perception can move toward one side of the Likert scale.

The construction of this form of analysis emerged in response to deficiencies found in pure descriptive statistics analyses, which would not be sufficient to capture nuances in ordinal scales. The analysis method indicates that, under conditions where the mean and standard deviation are identical, the Cns measure may differ due to the data spread within the scale. Thus, the mean and variability of these data, considering ordinal dispersion, are used to calculate the Cns value. Two main metrics can be employed: consensus and dissension measures (with the former being our focus). The use of a concept from thermodynamics known as Shannon entropy serves as an additional measure in the analysis of data dispersion and should be considered along with the Cns measure, as it evaluates the degree of disorder and irreversibility. The interpretation of the metrics complements each other as follows: the higher the Cns (agreement), the lower the entropy (degree of disorder). Similarly, the lower the Cns, the higher the entropy.

#### 2.4.3. Eye Movements

The visual inspection time was calculated from the moment that the stimulus was presented until the end of the stimulus presentation, as indicated by the participant, and the average in seconds was computed for each group of facial expressions presented.

The raw eye movement data, collected using the eye tracker, were recorded continuously throughout the experiment. Subsequently, the data were segmented based on each presentation period of the visual stimuli, corresponding to each of the 30 trials. These segmented records for each trial were then processed using the Eye Movements Metrics and Visualizations (EyeMMV) algorithm [48,49], implemented in Python version 3.10.11. This algorithm facilitated the identification of fixations and saccades within the eye movement data.

The EyeMMV algorithm is a dispersion-based method that employs spatiotemporal criteria, relying on two key parameters: tolerance (a spatial parameter) and the minimum duration threshold (a temporal parameter). We set the tolerance to 40 pixels, which corresponded to the projection of a circular area representing one degree of visual angle on the screen. This calculation took into account the participant’s distance from the screen. Additionally, we established the minimum duration threshold at 80 ms. These parameter choices were consistent with recommendations found in the literature for eye tracking methodologies and eye movement analysis [16].

Utilizing the fixation data, we derived additional metrics. The average number of fixations was determined by computing the total number of fixations from stimulus onset to completion and then dividing this by the number of times that each facial expression was presented under each condition. Similarly, the average fixation duration was calculated as the sum of the fixation durations divided by the number of stimulus presentations, providing an indication of cognitive function activity, with longer durations potentially reflecting heightened cognitive engagement [50].

## 3. Results

In this study, we examined three independent groups based on the level of restriction imposed by our experimental design on the participants’ visual field: no visual restriction (NVR), parafoveal and foveal vision (PFFV), and foveal vision (FV). The aim was to investigate the impacts of these experimental conditions on the visual inspection measures and the judgment of emotional facial expressions for five emotions (depicting happiness, sadness, anger, fear, and neutrality). Throughout the Section 3, the term “groups” refers to these independent variables (NVR, PFFV, and FV), unless explicitly mentioned. The following subsections detail our findings according to various scopes of analysis.

### 3.1. Summary Analysis by Experimental Conditions

The comparison of the overall accuracy among the experimental conditions using the Kruskal–Wallis test showed significant variations in recognition accuracy across groups ($\chi^{2}$ = 21.38, *p* = 0.000). The analysis of the effect size using the adjusted eta-squared indicated a moderate influence of the groups on the variability of the dependent variable ($\eta^{2}$ = 0.11). The post hoc Dunn’s tests highlighted notable differences, with the FV group differing with lesser accuracy than both the PFFV group (*p* = 0.002) and the NVR group (*p* = 0.000). Figure 2 summarizes the overall analysis by experimental condition, presenting violin plots for the NVR, PFFV, and FV groups, depicting various calculated measures. The inner part indicates the median with a white line and the quartile box in black, with the interquartile range highlighted in bold. The blue area represents the distribution density of the data. Statistical significance is denoted by “*” symbols.

### 3.2. Accuracy Variation across Expressions

The Kruskal–Wallis test indicated that the accuracy in recognizing facial expressions between groups was different for sadness ($\chi^{2}$ = 32.35, *p* = 0.000, $\eta^{2}$ = 0.18) and anger ($\chi^{2}$ = 16.17, *p* = 0.000, $\eta^{2}$ = 0.08). The post hoc Dunn’s tests also revealed that the FV group had significantly lower accuracy in recognizing sadness and anger expressions compared to the PFFV (*p* = 0.000) and NVR groups (p≤ 0.006). Descriptive statistics can be found in Table 1.

In conjunction with the previous analysis of the accuracy variation across expressions, we present the confusion matrices illustrating the distribution of emotional judgments in Figure 3. Each matrix corresponds to a different experimental condition: (a) NVR, (b) PFFV, and (c) FV. Within each matrix, rows represent the expected, or “correct”, emotional class, while columns correspond to the predicted emotion (judgment). The values in each cell indicate the proportion of each judgment, represented as a value between 0 and 1, where the sum of all judgments in a row equals 1.

Upon visually inspecting the main diagonal of the confusion matrices, it is evident that the FV condition exhibited the lowest accuracy across expressions, particularly for sadness. Furthermore, it indicated that sadness was frequently misinterpreted as either neutrality or fear, occurring in 18% and 12% of the judgments, respectively.

### 3.3. Confidence Analysis across Expression Judgments

The confidence of the judgments across expressions was assessed based on the participants’ confidence levels, which were measured using a Likert scale following their selection of the emotional expressions. This analysis involved calculating three complementary measures after converting the data into numerical values (see Section 2.4.2 for details of the methodological approach). These were the mean, entropy, and Cns.

In essence, the mean reflects the concentration of responses and indicates the overall confidence level around specific labels such as “Strongly agree” (SA), “Agree” (A), “Not sure” (N), “Disagree” (S), and “Strongly disagree” (SD). Please refer to Equation (Equation 3) for the corresponding numerical intervals. Entropy provides insight into the uncertainty in the distribution, complementing the information provided by the Cns. Finally, the Cns measure ranges from 0 to 1, representing the degree of consensus or agreement among participants, with 1 indicating total agreement and 0 representing complete disagreement (dissensus).

The comprehensive results of the analysis can be observed in Figure 4, where the mean, entropy, and Cns measures are plotted for better visualization. These outcomes reveal that the happiness expression presented the highest Cns for all experimental conditions, with values of 0.91, 0.85, and 0.87 for NVR, PFFV, and FV, respectively, indicating the greatest certainty in decision-making for this expression. Additionally, as expected, it exhibited the lowest entropy for all conditions, with the values of 0.51, 0.75, and 0.72, respectively.

In the NVR experimental condition, the lowest Cns values of 0.77 and 0.79 were observed for fear and neutrality, respectively, with corresponding entropy values of 1.12 and 1.08. For the PFFV condition, the lowest Cns values were 0.75 and 0.76 for neutrality and sadness, respectively, with corresponding entropy values of 1.32 and 1.25. For the FV condition, the lowest Cns values were 0.74 and 0.77 for sadness and neutrality, respectively, with corresponding entropy values of 1.39 and 1.23.

The neutrality expression exhibited the lowest Cns and mean values for the PFFV group and was closest to the lowest Cns and mean values within the NVR and FV groups. These values suggest that the participants in all experimental groups found it difficult to identify neutrality as an emotion with confidence, leading them to assess their judgments as less reliable, despite neutrality not being the least accurate in practice (see Figure 3).

The sadness expression exhibited the lowest consensus and mean values across all conditions and expressions, with Cns = 0.74 and mean = 0.70 in the FV group. This suggests a perception of uncertainty contributing to the highest level of disagreement among judgments. However, unlike the case with the neutrality expression, there was parity between the lowest confidence value and the lowest observed accuracy value in the judgments of sadness for the FV group.

Overall, the entropy increased for the two experimental conditions that imposed visual restriction compared to the NVR condition. This suggests a potential negative impact on the certainty of decision-making as the available visual angle for facial information extraction decreases. However, when analyzing the mean entropy for each group, the values were 0.91, 1.04, and 1.02 for NVR, PFFV, and FV, respectively, highlighting a significant difference compared to the NVR group but without an increase between PFFV and FV. From another perspective, when examining the variations in consensus between each group for a fixed expression, we observed a decrease in the Cns value for the expressions of happiness, sadness, and neutrality in the groups with visual restriction. Conversely, the expressions of fear and anger showed an increase in the Cns value for the groups with visual restriction.

It is evident that all expressions, in all conditions, had a predominantly high semantic concentration toward “Strongly agree” in their choices. Exceptions were observed for the neutrality expression in the PFFV condition and the sadness expression in the FV condition, where the resultant labels were “Agree”. It is worth noting that this pattern is directly observable in Figure 4, where the mean values can theoretically range from −1 to 1, yet all obtained measures fall within the positive interval of [0.70, 0.94].

Ultimately, apart from the semantic concentration information provided by the calculated mean, it is also noteworthy that the standard deviation, while typically indicating variability and potentially reflecting a degree of consensus, diverged significantly from the Cns values. This discrepancy can be attributed to the positively skewed distribution of the data, as indicated previously. This divergence between the standard deviation and Cns values suggests a nuanced understanding of the data’s variability and consensus. While the standard deviation may indicate a spread of responses, the Cns values offer insight into the overall agreement or consensus among participants. This discrepancy underscores the importance of considering multiple measures when interpreting the confidence and consensus of judgments in facial expression recognition tasks.

### 3.4. Eye Movement Analysis across Expressions

#### 3.4.1. Visual Inspection Time

The Kruskal–Wallis test identified noteworthy differences in the global visual inspection time across groups, yielding a significant result ($\chi^{2}$ = 108.58, *p* = 0.000). The effect size analysis suggested a substantial impact of the groups on the dependent variable ($\eta^{2}$ = 0.65). The post hoc Dunn’s tests highlighted differences between all groups (*p* = 0.000). Furthermore, the Kruskal–Wallis test demonstrated that the average inspection times for happiness ($\chi^{2}$ = 90.00, *p* = 0.000, $\eta^{2}$ = 0.54), sadness ($\chi^{2}$ = 106.71, *p* = 0.000, $\eta^{2}$ = 0.64), neutrality ($\chi^{2}$ = 105.01, *p* = 0.000, $\eta^{2}$ = 0.63), fear ($\chi^{2}$ = 79.65, *p* = 0.000, $\eta^{2}$ = 0.47), and anger ($\chi^{2}$ = 92.09, *p* = 0.000, $\eta^{2}$ = 0.55) expressions were statistically different among groups. These differences were observed in all groups for all expressions, as indicated by the post hoc Dunn’s tests (p≤ 0.013). Descriptive statistics for the inspection time can be found in Table 2.

#### 3.4.2. Fixation Metrics

The Kruskal–Wallis test revealed significant differences in the global number of fixations between groups ($\chi^{2}$ = 73.43, *p* = 0.000), with a large effect size ($\eta^{2}$ = 0.43). The pairwise post hoc Dunn’s tests showed that the number of fixations was higher in the FV group compared to both the NVR group (*p* = 0.000) and the PFFV group (*p* = 0.000). Furthermore, the Kruskal–Wallis test demonstrated that the average number of fixations differed significantly between groups for all expressions: happiness ($\chi^{2}$ = 59.46, *p* = 0.000, $\eta^{2}$ = 0.35), sadness ($\chi^{2}$ = 73.08, *p* = 0.000, $\eta^{2}$ = 0.43), neutrality ($\chi^{2}$ = 75.71, *p* = 0.000, $\eta^{2}$ = 0.44), fear ($\chi^{2}$ = 38.46, *p* = 0.000, $\eta^{2}$ = 0.21), and anger ($\chi^{2}$ = 53.66, *p* = 0.000, $\eta^{2}$ = 0.30). The post hoc Dunn’s tests revealed differences between the FV and NVR groups, as well as the PFFV and NVR groups, in the inspection of happiness, neutrality, and fear expressions (*p* = 0.000). Notably, there were differences between all groups in the inspection of sadness and anger expressions (p≤ 0.004).

In addition, there were significant differences in the fixation duration across groups, as indicated by the Kruskal–Wallis test ($\chi^{2}$ = 109.37, *p* = 0.000, $\eta^{2}$ = 0.66). The fixation duration increased with more visual restriction for FV/PFFV, FV/NVR, and PFFV/NVR (*p* = 0.000), as demonstrated by the pairwise post hoc Dunn’s tests. Furthermore, the Kruskal–Wallis test revealed that the average duration of fixations differed significantly between groups for all expressions: happiness ($\chi^{2}$ = 106.99, *p* = 0.000, $\eta^{2}$ = 0.64), sadness ($\chi^{2}$ = 108.70, *p* = 0.000, $\eta^{2}$ = 0.65), neutrality ($\chi^{2}$ = 107.03, *p* = 0.000, $\eta^{2}$ = 0.64), fear ($\chi^{2}$ = 103.69, *p* = 0.000, $\eta^{2}$ = 0.62), and anger ($\chi^{2}$ = 98.76, *p* = 0.000, $\eta^{2}$ = 0.59). The post hoc Dunn’s tests revealed differences between all groups. Notably, the duration of fixations increased progressively from NVR to FV (p≤ 0.000) in the inspection of happiness, sadness, neutrality, fear, and anger expressions. Descriptive statistics for the fixation metrics can be found in Table 3.

### 3.5. Impact of Facial Gender on Expression Recognition

Throughout this section, the analysis is constrained to reporting the influence of facial gender on expression recognition across the experimental conditions. To achieve this, the collected data were divided into subgroups based on the gender of the presented faces. In each subsection, a specific scope of analysis is delineated, with the respective measures of interest being the dependent variables. It is important to note that “M” and “IQR” in this section denote the median and interquartile range, respectively.

#### 3.5.1. Accuracy by Facial Gender

Upon examining the subset exclusively composed of male facial images, the comparison of the global accuracy between groups using the Kruskal–Wallis test showed significant variations in recognition accuracy among groups ($\chi^{2}$ = 7.60, *p* = 0.022). The analysis of the effect size using the adjusted eta-squared indicated a small influence of the groups on the variability of the dependent variable ($\eta^{2}$ = 0.02). The post hoc Dunn’s tests highlighted notable differences, with the FV group (M = 90, IQR = 80–93) showing lower accuracy than both the PFFV group (M = 93, IQR = 87–100) (*p* = 0.046) and the NVR group (M = 93, IQR = 87–100) (p≤ 0.050).

Correspondingly, upon examining the subset exclusively composed of female facial images, the comparison of the global accuracy between groups using the Kruskal–Wallis test showed significant variations in recognition accuracy among groups ($\chi^{2}$ = 23.28, *p* = 0.000). The analysis of the effect size using the adjusted eta-squared indicated a moderate influence of the groups on the variability of the dependent variable ($\eta^{2}$ = 0.12). The post hoc Dunn’s tests highlighted notable differences, with the FV (M = 80, IQR = 73–87) group showing lower accuracy than both the PFFV group (M = 93, IQR = 87–93) (*p* = 0.003) and the NVR group (M = 93, IQR = 87–100) (*p* = 0.000).

Figure 5 summarizes the overall analysis for the male and female facial genders by experimental condition. It presents split violin plots, where the blue violin represents the male gender and the orange represents the female gender, for the NVR, PFFV, and FV groups, depicting various calculated measures: (a) accuracy, (b) inspection time, (c) number of fixations, and (d) fixation duration. The inner lines indicate the median with a bold dotted line, and the interquartile range is highlighted by the other two dotted lines. The filled areas represent the distribution density of the data. Statistical significance is denoted by “*” symbols, following the same color scheme used for the violins.

#### 3.5.2. Expression Judgment by Facial Gender

In the male facial image subset, the Kruskal–Wallis test indicated that the accuracy between groups was different for sadness, with a moderate effect ($\chi^{2}$ = 15.71, *p* = 0.000, $\eta^{2}$ = 0.07). The post hoc Dunn’s tests also revealed that the FV group (M = 67, IQR = 33–100) had significantly lower accuracy in recognizing sadness expressions than the PFFV (M = 100, IQR = 67–100) (*p* = 0.001) and NVR groups (M = 100, IQR = 67–100) (*p* = 0.005).

In contrast, in the female facial image subset, the Kruskal–Wallis test indicated that the accuracy between groups was different for sadness, with a large effect size ($\chi^{2}$ = 31.37, *p* = 0.000, $\eta^{2}$ = 0.17), and anger, with a moderate effect size ($\chi^{2}$ = 17.28, *p* = 0.000, $\eta^{2}$ = 0.08). The post hoc Dunn’s tests also revealed that the FV group had significantly lower accuracy in recognizing sadness (M = 67, IQR = 33–100) and anger expressions (M = 67, IQR = 67–100) than that seen for the sadness (M = 100, IQR = 67–100) and anger expressions (M = 100, IQR = 100–100) presented in the PFFV and NVR groups (both expressions with M = 100, IQR = 100–100) (p≤ 0.001).

Additionally, Figure 6 complements the preceding analysis of expression judgment by facial gender across the experimental conditions. It illustrates the combined confusion matrices depicting the distribution of emotional judgments for the male and female facial genders. Each matrix corresponds to a different experimental condition: (a) NVR, (b) PFFV, and (c) FV. Within each matrix, rows represent the expected, or “correct”, emotional class, while columns correspond to the predicted emotion (judgment). In each matrix cell, the upper triangle indicates values for male faces, and the lower triangle indicates values for female faces. These values denote the proportion of each judgment over the participants, represented as a value between 0 and 1, where the sum of all judgments in a row equals 1.

The Wilcoxon test results revealed that female facial expressions (M = 93, IQR = 87–93) were recognized with greater accuracy than male facial expressions (M = 93, IQR = 87–100) in the PFFV condition (*W* = 229.50 with *p* = 0.013 and ρ = 0.87). In contrast, male facial expressions (M = 90, IQR = 80–93) were recognized with greater accuracy than female facial expressions (M = 80, IQR = 73–87) in the FV condition (*W* = 254.50 with *p* = 0.013 and ρ = 0.80).

#### 3.5.3. Visual Inspection Time by Facial Gender

Within the subset of male facial images, the Kruskal–Wallis test identified significant differences in the global visual inspection time across groups, yielding a significant result ($\chi^{2}$ = 106.89, *p* = 0.000). The effect size analysis suggested a large effect of the groups on the dependent variable ($\eta^{2}$ = 0.63). The post hoc Dunn’s tests highlighted differences between all groups (*p* = 0.000), FV (M = 19.40, IQR = 11.71–27.15), PFFV (M = 11.07, IQR = 7.72–14.81), and NVR (M = 4.32, IQR = 2.46–5.67).

Similarly, within the subset of female facial images, the Kruskal–Wallis test identified significant differences in the global visual inspection time across groups, yielding a significant result ($\chi^{2}$ = 107.59, *p* = 0.000). The effect size analysis suggested a large effect of the groups on the dependent variable ($\eta^{2}$ = 0.65). The post hoc Dunn’s tests highlighted differences between all groups (*p* = 0.000) FV (M = 20.16, IQR = 14.72–27.36), PFFV (M = 10.76, IQR = 8.56–15.26), and NVR (M = 4.73, IQR = 2.82–5.91).

The Wilcoxon test results revealed that female facial expressions were inspected with more time than male facial expressions in the FV condition (*W* = 844.00 with *p* = 0.046 and ρ = 0.34).

#### 3.5.4. Fixation Metrics by Facial Gender

Among the male facial image subset, the Kruskal–Wallis test analysis revealed significant differences in the global number of fixations between groups ($\chi^{2}$ = 70.65, *p* = 0.000), with a large effect size ($\eta^{2}$ = 0.42). The pairwise post hoc Dunn’s tests showed that the number of fixations was higher in the FV group (M = 20.33, IQR = 14.95–29.70) compared to the NVR group (M = 8.33, IQR = 5.35–10.88) (*p* = 0.000) and the PFFV group (M = 16.80, IQR = 12.26–21.53) compared to NVR (*p* = 0.000). Additionally, there were significant differences in the fixation duration across groups, as indicated by the Kruskal–Wallis test ($\chi^{2}$ = 108.11, *p* = 0.000, $\eta^{2}$ = 0.65). The fixation duration increased with more visual restriction in the FV/PFFV, FV/NVR, and PFFV/NVR comparisons (FV M = 792.82, IQR = 692.73–855.32; PFFV M = 513.25, IQR = 481.15–567.22; NVR M = 359.26, IQR = 324.38–436.33), as demonstrated by the pairwise post hoc Dunn’s tests (*p* = 0.000).

Similarly, among the female facial image subset, the Kruskal–Wallis test analysis revealed significant differences in the global number of fixations between groups ($\chi^{2}$ = 74.28, *p* = 0.000), with a large effect size ($\eta^{2}$ = 0.44). The pairwise post hoc Dunn’s tests showed that the number of fixations was higher in the FV group (M = 21.13, IQR = 16.55–29.18) compared to both the NVR group (M = 9.03, IQR = 5.16–11.26) (*p* = 0.000) and the PFFV group (M = 16.00, IQR = 13.26–21.66) (*p* = 0.038), and also differed between the PFFV and NVR groups (*p* = 0.000). Moreover, there were significant differences in the fixation duration across groups, as indicated by the Kruskal–Wallis test ($\chi^{2}$ = 107.12, *p* = 0.000, $\eta^{2}$ = 0.64). The fixation duration increased with more visual restriction in the FV/PFFV, FV/NVR, and PFFV/NVR comparisons (FV M = 808.73, IQR = 674.09–884.19; PFFV M = 524.85, IQR = 488.05–596.28; NVR M = 356.09, IQR = 322.45–424.57), as demonstrated by the pairwise post hoc Dunn’s tests (*p* = 0.000).

The Wilcoxon test revealed no difference between the number of fixations for female facial expressions and male facial expressions in any condition. Similarly, the fixation duration exhibited the same tendency.

## 4. Discussion

In this study, we explored the influence of visual field restriction conditions—especially FV and PFFV, in comparison to NVR—on basic emotional facial expression recognition. Here, for ease of reference, we simply refer to them as visual conditions. The investigation encompassed various facets of expression recognition, including accuracy, consensus, the inspection time, the number of fixations, and the fixation duration across distinct conditions. The results unveil a nuanced relationship between visual perceptual and attentional processes in recognizing emotional facial expressions.

Our findings challenge the conventional emphasis on foveal vision in facial perception studies, revealing the pivotal role of parafoveal vision, which encompasses both the foveal and parafoveal regions, in accurate expression recognition. Notably, recognition accuracy, inspection time, number of fixations, and fixation duration differences emerged across the groups in the different visual conditions. First, the visual condition impacted the global recognition accuracy. The FV had lower accuracy than both the PFFV and NVR conditions, underscoring the importance of parafoveal and peripheral vision in expression recognition. Meanwhile, no significant difference was observed in accuracy between the PFFV and NVR conditions, suggesting that the PFFV is sufficient for accurate expression recognition. Second, we found that the inspection times varied significantly among the groups. The participants spent different amounts of time inspecting facial expressions based on the visual condition, suggesting that attention allocation differs under these conditions. Third, the number of fixations increased as the level of visual restriction increased, indicating that the participants performed more fixations and spent more time fixating on the face when their vision was constrained.

Regarding accuracy, the recognition of specific emotions, especially sadness, anger, and neutrality, notably decreased in the FV group compared to other experimental conditions. The analysis of consensus (Cns) and entropy provided valuable perspectives on the participants’ agreement levels and the disorder in their decision-making processes. Notably, the consensus varied across expressions and experimental conditions. In all experimental conditions, there was higher consensus and lower entropy for happiness, reinforcing the happy face advantage [9,51,52,53]. There was less consensus and higher entropy for sadness, followed by neutrality in FV and PFFV. This suggests that these expressions may require enhanced processing at both the peripheral and central vision levels, differentiating them from other facial expressions. Our findings align with prior research by Bombari et al. [54], emphasizing that the recognition of sadness relies on configural information. Despite the methodological differences, both studies agree on the increased difficulty in identifying scrambled sad faces, in contrast to intact or blurred ones, underscoring the essential role of capturing the interrelationships between facial features for accurate recognition. However, conflicting findings exist regarding the crucial areas for the recognition of these expressions [7,55].

Although there was no similar impact on the accuracy measure, the Cns measure for anger was rated with more consensus in PFFV and FV. These findings align with a recent study conducted by Atkinson and Smithson [20], where specific areas of emotional faces located in the foveal field, processed briefly within a single fixation, were found to enhance expression recognition compared to areas processed in the extrafoveal region. However, it is important to note that while their study observed the better recognition of angry faces within the foveal area, the results in the present study do not entirely replicate this finding considering the significant loss in accuracy presented in the FV group. Notably, observations by Poncet et al. [53] and Bombari et al. [54] identified fixations on the centers of angry faces. These collective outcomes support the hypothesis that expressions of anger may engage multiple areas of interest (AOI) compared to other facial expressions [56], suggesting potential holistic processing [54], thereby reinforcing the findings of this study.

Perceiving neutral faces poses significant challenges. It can be influenced by cultural differences, racial stereotypes, and individual expectations, with a profound impact of the surrounding context [57,58,59]. Contrary to the traditional view of neutral faces as being devoid of emotional content, studies indicate that they can convey strong messages when paired with specific emotional–contextual information [60]. Contextual factors, such as situational cues from external sources (e.g., other emotional faces for comparison or contextual background), are well established to significantly impact the interpretation of neutral facial expressions [61]. Moreover, Suess et al. [62] demonstrated that an individual’s affective knowledge can also influence their perception of facial expressions, including those considered neutral, suggesting a susceptibility to external influences such as personal experiences, interpersonal relationships, or personality traits. In this context, our results indicated the lowest or the closest to the lowest consensus and confidence levels for neutrality judgment among all groups, which is consistent with the previous literature. This divergence may have been potentiated by the proximal interference of other facial expressions under judgment, acting as emotional contexts. However, the same extent of effect was not observed in the accuracy.

Furthermore, significant differences in inspection times among groups were observed, with an increase corresponding to the level of visual restriction. This processing trend aligns with the findings from a study on identity recognition by Maw and Pomplun [29], indicating that the accuracy decreases with visual restriction conditions while the response time increases, despite variations in the visual angles employed compared to our study. Notably, in the present study, PFFV exhibited a sufficient inspection time to equalize the accuracy to NVR, in contrast to the FV condition. The disparities were most evident in the recognition of sadness and anger across all conditions, suggesting that these expressions are especially responsive to variations in visual conditions. Unlike other expressions, the differences were observed only between the FV and NVR conditions. It is noteworthy that the accuracy results of this study do not align with those of Kim et al. [21]. However, it is important to consider that their study solely utilized a mouse-tracking paradigm, which could have introduced variations compared to our eye tracking control [26]. Additionally, the specific visual angle in degrees was not clearly specified in the study, potentially contributing to the divergent findings.

Substantial variations in the mean number of fixations and fixation duration were observed among the groups, with visual restriction leading to an increase in both measures. This pattern aligns with studies employing similar methodologies, such as investigations into face identity recognition [29] and face inversion effects [28]. For instance, Kreichman et al. [63] demonstrated that the impact of eccentricity (the distance of a stimulus from the point of fixation) on the recognition of face images differed from its impact on house images. Specifically, their study found that faces were more affected by eccentricity, leading to reduced accuracy in face discrimination tasks compared to house discrimination tasks at greater eccentricity. This finding suggests that the ability to perceive and recognize faces declines more rapidly with increasing eccentricity. However, it is important to note that the methodologies employed in their study differed from ours.

The study of Fiorentini [64] observed a time increase in processing as the number of items increased, predominantly impacting foveal and parafoveal vision. The author discussed the distinct properties of fovea and parafovea pattern processing during visual search, highlighting differences in size and the spatial frequency spectrum, and, according to the time course, they observed different temporal profiles among the visual responses. Specifically, fine elements viewed foveally could only be searched serially, whereas larger elements projected on an area immediately surrounding the fovea could be searched in parallel. We posit that the increase in processing observed in our study, and the accuracy impact in the foveal condition, aligns with this presupposition. Considering the complexity of the face, from 5° in PFFV, individuals demonstrated similar accuracy to NVR, possibly indicating the parallel processing of information, albeit with an increased processing time. Neurophysiological studies suggest that the visual processing of faces can vary between the fovea and parafovea, potentially due to differences in the attention-related sensory gain control operating in these regions [65].

In the analysis segmented by the gender of the faces, it was observed that both male and female facial expressions were recognized more accurately under the NVR and PFFV conditions compared to FV. Furthermore, the inspection time for both male and female expressions varied across all conditions, increasing with the degree of visual restriction. The number of fixations on male facial expressions differed between the NVR condition compared to PFFV and FV, while female expressions showed differences across all conditions. Moreover, male and female expressions exhibited differences in duration across all experimental conditions. In summary, these analyses suggest that at a 5° visual angle (PFFV), the accuracy tends to approach that of the unrestricted visual field, albeit with a greater demand for inspection time and fixations in both quantity and duration. Notably, male facial expressions showed no differences in the number of fixations between the FV and PFFV conditions, implying similarity in this metric for the recognition of male facial expressions. This observation suggests that achieving similar accuracy to the NVR condition in the PFFV condition may require as many gaze allocations as in the FV condition. These findings contribute to advancements in understanding how facial expressions are recognized based on facial gender [11].

Few studies have investigated the processing at the level of the parafovea and fovea in recognizing facial expressions. This is the first to restrict the visual field to these areas in order to study their impact on the recognition of basic human facial expressions. These findings have practical implications for various fields, including clinical psychology and human–computer interaction. Understanding the influence of visual conditions on facial expression recognition can guide diagnostic approaches and behavioral therapeutic interventions [66]. Moreover, these insights may inform the development of algorithms for facial recognition technology, which can benefit from a better understanding of human visual perception. In essence, this study advances our understanding of perceptual and attentional cognitive processes in the context of recognizing emotional facial expressions. The complex interaction between visual conditions and fixation patterns underscores the need to consider parafoveal and peripheral vision, expanding our comprehension of the mechanisms behind this fundamental aspect of human communication and social interaction.

Nevertheless, the limitations of this study mainly arise from implications for the determined criteria used to restrict the visual field for parafoveal vision. Firstly, the choice to examine the PFFV with a $5^{{}^{\circ}}$ aperture limited our understanding of the specific characteristics of the visual processing within the parafoveal extension ($2^{{}^{\circ}}$ to $5^{{}^{\circ}}$). The fixed $5^{{}^{\circ}}$ PFFV aperture hindered the evaluation of a more precise point of saturation for the observed differences in accuracy between the experimental conditions. Secondly, it is important to acknowledge that the observed contributions of PFFV in this study included influences from parafoveal and foveal information simultaneously, thus limiting the understanding of parafoveal vision contributions alone. For example, it was not possible to ascertain whether the observable changes were mostly due to the parafoveal contribution or occurred only when the fovea and parafovea were simultaneously engaged.

Future research could address the first limitation by exploring intermediate levels of PFFV restriction to assess their impacts on the accuracy and to identify the saturation point more precisely, at which the accuracy stabilizes and no longer increases. This would allow for a more comprehensive understanding of the effects of the PFFV interplay according to the increase in visual field aperture and its influence on perceptual processes. Regarding the second limitation, isolating the contributions of parafoveal vision could be beneficial to assess the interaction dynamics between the parafovea and fovea more clearly. This could be achieved by setting specific conditions for isolated parafoveal vision (e.g., central occlusion with $2^{{}^{\circ}}$ to $5^{{}^{\circ}}$ aperture) to independently evaluate its role and interaction with foveal vision. In doing so, we could enhance our understanding of how these different visual components contribute to visual perception and their interplay mechanisms.

It is important to note that this study did not analyze the gaze distribution across areas of interest. Instead, we focused on analyzing metrics related to the perception and processing of emotional facial expressions, without making prior assumptions about the importance of specific image areas across experimental conditions. We aimed to assess the impact of visual field restriction on expression recognition and its significance, prioritizing the statistical analysis of the available metrics. In our judgment, introducing spatial gaze data analysis at this stage would have detracted from the study’s primary focus, potentially confounding the main objectives and findings. Furthermore, given the experimental design and methodological constraints imposed by the visual field restriction and free inspection time, we recognize that interpreting gaze distribution data could have been challenging and extensive. This warrants exploration in a separate study, focusing on differences related to the visual processing strategies.

Overall, our study reveals a crucial finding regarding the influence of visual field restriction on facial expression recognition. Specifically, the significant differences among the NVR/FV and PFFV/FV groups in accuracy, despite no difference within NVR/PFFV, underscore the impact of parafoveal vision on facial expression recognition, highlighting its importance in the visual processing of emotional faces. As a practical implication, our results suggest that eye tracking data analysis methods should incorporate projection angles extending to at least the parafoveal level. Additionally, these potential methodological adaptations could help to clarify the results of various studies on facial emotion processing, facilitating the establishment of consensus. Future research should build upon these findings by investigating the impact of visual restrictions on specific facial areas of interest, thereby contributing to a more comprehensive understanding of facial expression recognition.

## 5. Conclusions

In conclusion, our study delved into the intricate dynamics of visual field restriction conditions, specifically foveal and parafoveal vision, versus no visual restriction, on the recognition of basic emotional facial expressions. Contrary to the traditional emphasis on foveal vision, our findings underscored the critical role of parafoveal vision inclusion in accurate expression recognition, with significant differences in accuracy, inspection times, the number of fixations, and durations across conditions. Notably, sadness, anger, and neutrality recognition were particularly affected in the foveal group, challenging conventional wisdom. The consensus and entropy analysis revealed expression-specific variations, emphasizing the complexity of the decision-making processes. Our findings suggest that incorporating projection angles extending to at least the parafoveal level could improve the eye tracking data analysis methods, clarifying the results in facial emotion processing studies, and aiding consensus-building in the field. Our research, the first to evaluate foveal and parafoveal vision in studying basic human facial expressions, carries implications for clinical psychology and human–computer interaction, offering valuable insights for diagnostic and therapeutic purposes, as well as the development of facial recognition technology algorithms. In sum, our findings advance our understanding of perceptual and attentional processes in facial expression recognition, highlighting the need to consider parafoveal and peripheral vision for a comprehensive grasp of human communication dynamics.

## Figures and Tables

**Figure 1 behavsci-14-00355-f001:**
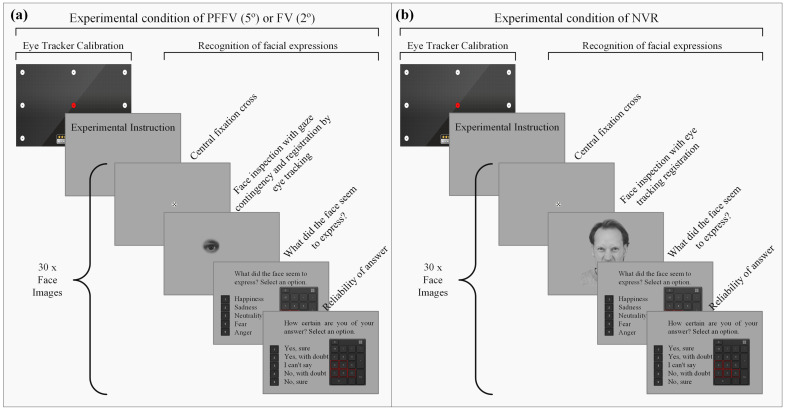
Experimental design of collection procedure according to experimental condition. The screen order is presented from top to bottom. The group of screens presenting the random facial stimulus and its judgment is repeated 30 times. (**a**) Experimental groups PFFV and FV differ only in the aperture size for the moving window controlled by the participants’ gaze detected by the eye tracker. (**b**) Experimental group NVR presents the full image. The facial image in the example was obtained from the KDEF image database (ID: AM10ANS).

**Figure 2 behavsci-14-00355-f002:**
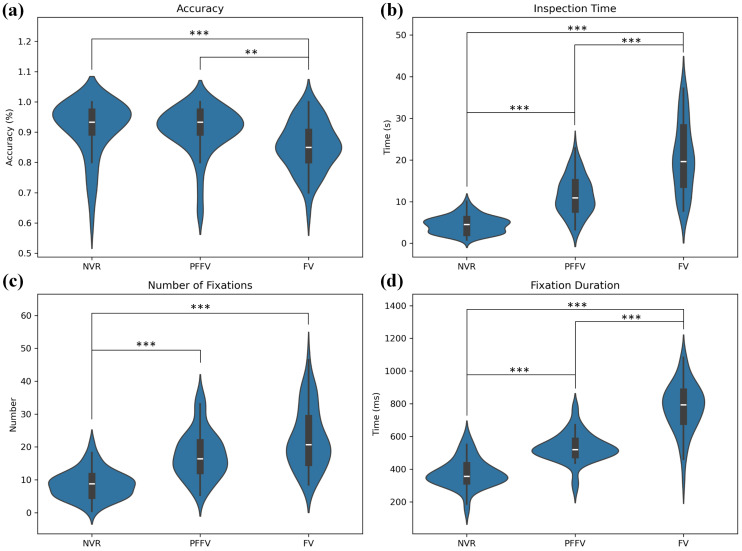
Summary analysis by experimental condition. Violin plots for (**a**) accuracy, (**b**) visual inspection time, (**c**) number of fixations, (**d**) fixation duration. Note: **, and *** denote statistical significance at the p<0.01, and p<0.001 levels, respectively.

**Figure 3 behavsci-14-00355-f003:**
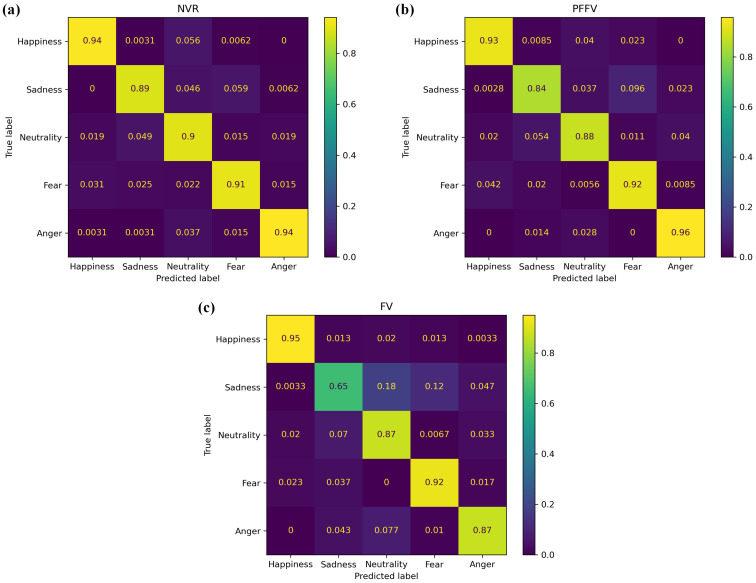
Confusion matrix illustrating the distribution of emotional judgments for facial emotional expressions across different conditions: (**a**) NVR, (**b**) PFFV, and (**c**) FV.

**Figure 4 behavsci-14-00355-f004:**
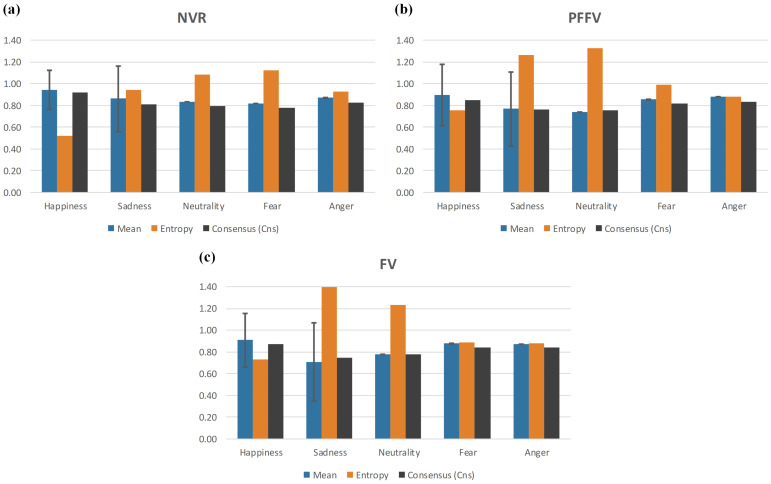
Confidence in expression judgment. Mean, entropy, and Cns in recognizing facial emotional expressions across experimental conditions: (**a**) NVR. (**b**) PFFV. (**c**) FV. Note: The error bar indicates the standard deviation over the mean.

**Figure 5 behavsci-14-00355-f005:**
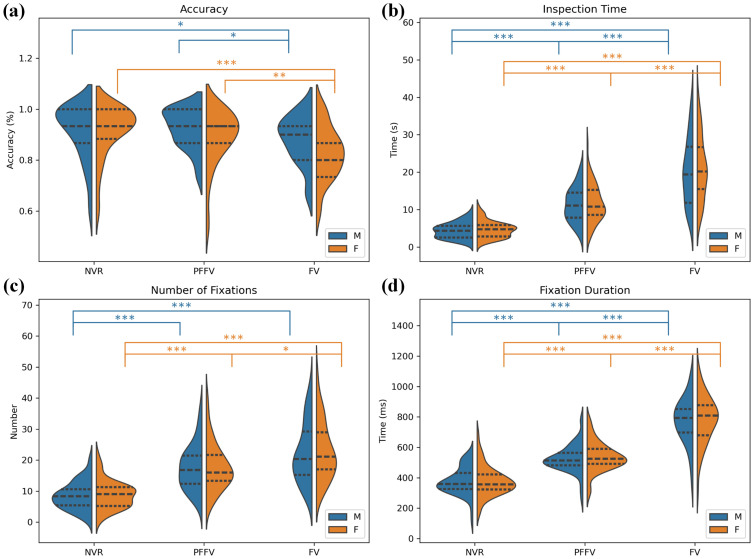
Summary analysis of facial gender impact across experimental conditions. Violin plots for (**a**) accuracy, (**b**) visual inspection time, (**c**) number of fixations, (**d**) fixation duration. The blue color represents the male gender, and the orange color represents the female gender. Note: *, **, and *** denote statistical significance at the p≤0.05, p<0.01, and p<0.001 levels, respectively.

**Figure 6 behavsci-14-00355-f006:**
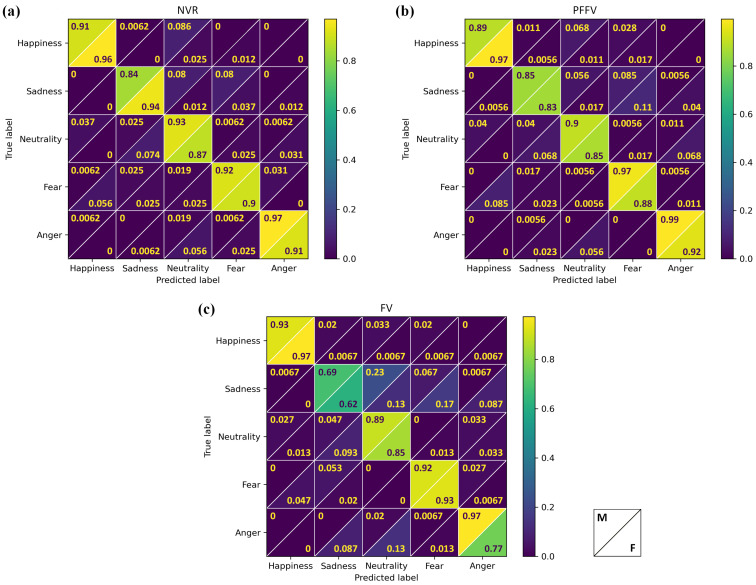
Confusion matrix illustrating the distribution of emotional judgments for male and female facial gender across experimental conditions: (**a**) NVR, (**b**) PFFV, and (**c**) FV. For each matrix cell, the upper triangle indicates values for male faces, and the lower triangle indicates values for female faces.

**Table 1 behavsci-14-00355-t001:** Accuracy in recognizing facial emotional expressions across different conditions.

Expression	NVR	PFFV	FV
Happiness	100 (96–100)	100 (83–100)	100 (100–100)
Sadness	100 (83–100)	83 (83–100)	67 (50–83)
Neutrality	100 (83–100)	83 (83–100)	83 (83–100)
Fear	100 (83–100)	100 (83–100)	100 (83–100)
Anger	100 (83–100)	100 (100–100)	83 (83–100)
Overall	93 (90–97)	93 (90–97)	85 (80–90)

Median followed by the interquartile range in parentheses.

**Table 2 behavsci-14-00355-t002:** Visual inspection time in recognizing facial emotional expressions across different conditions.

Expression	NVR	PFFV	FV
Happiness	3.14 (2.21–4.52)	9.51 (6.30–13.44)	14.78 (9.13–23.91)
Sadness	4.39 (2.73–6.20)	10.99 (9.06–14.05)	23.46 (17.46–35.55)
Neutrality	4.52 (2.53–7.16)	13.66 (9.01–19.44)	24.71 (16.90–29.74)
Fear	3.94 (2.42–6.23)	8.74 (6.79–15.21)	13.87 (10.27–21.11)
Anger	3.96 (2.36–5.99)	8.62 (5.93–12.79)	18.44 (11.73–25.30)
Overall	4.51 (2.46–5.85)	10.90 (8.14–14.68)	19.64 (13.31–28.18)

Median followed by the interquartile range in parentheses.

**Table 3 behavsci-14-00355-t003:** Number of fixations and fixation duration (ms) in recognizing facial emotional expressions.

Measurement	Expression	NVR	PFFV	FV
Number of Fixations	Happiness	6.16 (4.75–8.83)	14.83 (10.00–21.66)	15.83 (11.41–24.75)
	Sadness	8.66 (5.50–11.04)	16.66 (13.33–21.66)	26.08 (18.50–37.54)
	Neutrality	9.16 (5.29–14.25)	19.83 (15.00–28.66)	25.91 (18.08–33.75)
	Fear	6.83 (4.79–12.58)	14.16 (10.33–23.50)	17.25 (11.08–24.41)
	Anger	7.16 (4.45–10.54)	12.66 (9.16–18.00)	19.16 (13.29–25.50)
	Overall	8.81 (5.20–11.18)	16.40 (12.76–21.40)	20.73 (15.07–29.03)
Fixation Duration	Happiness	364.37 (320.22–410.75)	508.62 (451.00–551.91)	759.87 (676.42–861.14)
	Sadness	364.89 (333.42–421.85)	544.34 (508.48–604.70)	844.00 (704.06–927.63)
	Neutrality	344.41 (320.83–421.90)	522.76 (483.50–573.04)	793.59 (661.12–889.84)
	Fear	344.41 (311.52–453.87)	509.86 (478.49–567.83)	755.54 (674.09–836.89)
	Anger	357.53 (308.56–451.49)	515.16 (485.28–555.01)	811.61 (697.99–901.52)
	Overall	356.48 (326.08–424.76)	520.08 (487.31–572.60)	794.15 (688.19–878.52)

Median followed by the interquartile range in parentheses.

## Data Availability

The data underlying the results presented in the study are available at https://doi.org/10.5281/zenodo.10703513. The source code for the experimental data collection software is available in a GitHub repository at https://github.com/melinaurtado/VisFieldFacialExpCollect and archived in Zenodo with DOI https://doi.org/10.5281/zenodo.10700824. The source code for the data analysis software is available in a GitHub repository at https://github.com/rafaeldr/VisFieldRestrictAnalysis and also archived in Zenodo with DOI https://doi.org/10.5281/zenodo.10711813. All other relevant data are included within the manuscript.

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
