# Peer review of "Visual Field Restriction in the Recognition of Basic Facial Expressions: A Combined Eye Tracking and Gaze Contingency Study"

_behavsci, 2024, doi:10.3390/bs14050355_

Round 1

Reviewer 1 Report

Comments and Suggestions for Authors

The study claims to have isolated parafoveal vision (PFV) and foveal vision (FV) in expression recognition for the first time. The results reveal accuracy differences between FV, PFV, and NRV groups, underscoring parafoveal vision's importance. That sounds very innovative. But there are some problems that need to be introduced. (1) The biggest problem is that the experimental design and process introduction are not detailed, especially how to realize the separation of PFV and FV in expression recognition. The left graph of Figure 1 shows only one independent variable level, but the heading is PFV and FV, which is confusing. In the introduction, the authors say that they combine two technologies, such as Moving Window Technique and eye tracking, so how do these two technologies combine, and why do they combine to achieve separation above?  (2) In the introduction or discussion section, the authors should detail the differences in experimental procedures between this study and existing PFV studies, so that readers can judge the innovation of this study. (3) Previous studies have found that FV is very important in expression recognition, but this study found PFV is more important. The authors need to analyze the reasons for the inconsistencies from the experimental procedures and possible mechanisms. (4) The contribution of FV and PFV in expression recognition can be further calculated and separated by using the data of this study.

Author Response

Q1 RESPONSE: We have carefully reviewed and adjusted the entire manuscript to ensure that the research scope is now clear and well-delimited throughout the relevant sections. To disambiguate and make a clear distinction between the experimental conditions and independent variables, we have revised the old terminology "PFV" to "PFFV" (parafoveal and foveal vision), while maintaining the others: FV (foveal vision) and NVR (no visual restriction). The previous terminology was, in fact, contributing to various misconceptions for the readers. For the same reason, we also removed terms like "isolated". Our study investigates the role of visual field restriction up to the level of the parafovea, wherein the PFFV condition, the projection of the visual angle is wider to include the parafoveal vision together with foveal vision.

         Regarding the left graph of Figure 1, it indicates the experimental procedure for both PFFV and FV conditions, since they differ only in the aperture size for the moving window. However, we have now provided clear indications for this in the Figure caption and the related sections of the manuscript (see lines 183-186).

         Regarding the combination of the moving window technique and eye tracking, we have provided a better description in the section "2.3. Procedure":

         Lines 187-190: "The visual restriction was implemented using the moving window technique, directly controlled by the eye tracker in response to the viewer’s eye gaze. A mask was applied over the stimulus, with a visual aperture diameter corresponding to the projection of the respective visual angle on the screen, 5â—¦ for PFFV or 2â—¦ for FV."

            Thank you for bringing these points to our attention, and we believe these changes will improve the overall clarity and coherence of the manuscript.

Q2 RESPONSE: We have revised the introduction and discussion sections to incorporate the differences in experimental procedures between our study and related research on facial emotion recognition that applies gaze contingency. Furthermore, we have provided clearer explanations of our research scope and target evaluation task, specifically addressing the misconceptions resulting from the previous flaws in terminology, as raised in question 1.

Q3 RESPONSE: Indeed, foveal vision (FV) is of significant importance, and our previous terminology may have misled the interpretation of our findings. What our study demonstrates is that including sufficient visual angle projection to also encompass parafoveal vision (PFFV) leads to a significant increase in accuracy compared to foveal vision (FV) alone. Complementarily, unrestricted vision (NVR) did not show a significant increase in accuracy compared to the PFFV group.

         In the context of our findings, the inconsistencies observed in various studies may be, at least in part, related to the utilization of analysis parameters that predominantly prioritize the examination of foveal vision angle, neglecting potential influences from simultaneous parafoveal and peripheral information.

            We have made efforts to adjust the relevant sections where this information is crucial, ensuring clarity and accuracy in our presentation.

Q4 RESPONSE: We have adjusted the terminologies in the results section to clarify the outcomes derived from the differentiation among the experimental groups NVR (no visual restriction), PFFV (parafoveal and foveal vision), and FV (foveal vision). Our analyses encompass a summary analysis by experimental condition, an analysis of accuracy variation across expressions, a confidence analysis across expressions judgment, an eye movement analysis across expressions, and the impact of facial gender on expression recognition. This latter one was included in the review process.

Reviewer 2 Report

Comments and Suggestions for Authors

1) Title and Line 1: The expressions “Exploring Visual inspection” and Understanding Visual Inspection” are meaningless. Possibly “Exploring the role of…”, or similar? Also, “Visual inspection” per se is not clearly defined, see also line 26: “visual inspection varies…”: what is it exactly that varies? are the Authors referring to numbers of inspected areas? To which specific areas are inspected? To the duration of the inspection? …? only much later in the paper are criteria defined, they should be mentioned earlier in the text

2) I see several problems with the methods and the way they are described:

First of all, information on the specific regions examined during the visual inspection are missing, although several previous studies are cited where this piece of information is provided and deemed relevant. Looking at Figure 1, it would appear as only a small part of the face is presented.

Related to this first question, is the fact that according to the protocol what is called “parafoveal vision” appears to be both foveal AND parafoveal, in that the fovea is not masked: is this correct? this fact changes substantially the considerations that can be made discussing the findings

Also:

Line 111: How were the participants recruited? were they all university students? did they get a fee or compensation for their participation?

Line 140: The reason for the choice of the four expressions is not explained

Line 144: by which criteria are the models “similar” to the participants? how about sex?

Lines 181 and following: how exactly “the eye tracking induced visual field restriction” is never described

Lines 186-187: contradictory sentences: why free fixation time, if it has been demonstrated that two fixations are enough?

Line 214: what’s the definition of “weighted” mean as referred to the Likert scale? “weighted” by which parameter? Also, what’s the advantage of considering CNS and Entropy over just a mean? Later, in Figure 2, the curve of CNS and Mean largely overlap

Line 245: what do the Authors mean by semantic counterpart?

Line 264: the sentence “two dispersion parameters within a circular area of one degree” is obscure, please explain.

Line 266: What is the “temporal parameter”? does it mean “duration”? i.e., a minimum duration of 100 ms is required to identify a fixation period?

Line 271-275: The procedure described in lines 271-274 does not seem to warrant the conclusion at line 275, in that areas of the face are never mentioned in lines 271-274.

Line 281: the term “increased cognitive function” is ambiguous: do the Authors mean increased cognitive effort? Increased cognitive efficiency? Performance? …Other? Please clarify.

Line 284: Facial expressions cannot be considered as dependent variables

Line 290 (and later, line 298): what is “the dependent variable”?

3) Also, the description of the results is generally not very clear

Lines 317-318: “In the NVR experimental condition, we observed lower Cns = 0.77 and Entropy = 0.95 for fear and neutrality, with Cns = 0.78 and Entropy = 1.02, respectively.” is incomprehensible, please clarify

Line 326-328: This sentence needs explanation: what is the semantic differential, from which data is it evident where the semantic differential is located?

Figure 2 categorical variables should not be represented with continuous curves but with histograms; the tables below each graph are not necessary, std deviation marks should be added to the histograms

Tables: significance asterisks could be added, but even better, the tables should all be replaced by figures (histograms), with significance asterisks; also, units should always be specified

Minor observations:

Throughout the paper, the facial expressions are variably referred to by either nouns or adjectives, please be consistent: e.g., “sad, fear” should be either “sadness, fear”, or “sad, fearful”

Line 5: NRV or NVR? mixed throughout the paper

Line 9: explain MMV

Line 9: Why preliminary results??

Line 69 and elsewhere: Why is the future tense (=“will”) used? The study has been performed already

Lines 192-193: responses texts are different from what appears in the figure

Line 202 “what’s the meaning of “and the average for each expression category”?

Lines 269-270: The sentence “were evaluated; with a saccade representing the interval between fixations” is obscure, maybe “;” should be “,”?

Line 385: “layer of insights”??

Line 399: “the consensus measure for anger was rated with more consensus”: the syntax is flawed

Line 433: “differential modulations of faces and houses”: what does it mean? modulation of what? location? recognition?

Line 437: “and observing through temporal profiles of visual responses”: meaning?

Comments on the Quality of English Language

In several sentences there are grammar and/or syntax flaws, see "minor observations" above

Author Response

Q1 RESPONSE: We have revised the title of the manuscript to better reflect the content and purpose of the study, incorporating your suggestion to enhance clarity. Additionally, we have carefully reviewed and clarified the mentions of "visual inspection" throughout the manuscript, ensuring that they are appropriately contextualized and defined. Thank you for bringing these points to our attention, and we believe these changes will improve the overall clarity and coherence of the manuscript.

Q2 RESPONSE: We have carefully reviewed and adjusted the manuscript to ensure that the research scope is clear and well-delimitated. In response to your feedback, we have made adjustments to Figure 1 description and caption to accurately indicate the experimental conditions and procedures. Additionally, we acknowledge your observation regarding the terminology used to describe parafoveal experimental conditions, and you are correct that our previous terminology may have been misleading. We have revised the terminology to "PFFV" (parafoveal and foveal vision) to accurately reflect the experimental design. Furthermore, we have removed terms such as "isolated" when referring to distinct levels of visual restriction, as they may have conveyed an inaccurate impression to the reader.

         Regarding the clarification of regions (or areas) of interest in our manuscript. We acknowledge the importance of addressing the divergence in the literature as motivation for evaluating the possible influence of the field of view size on eye tracker data analysis and its contribution to these discrepancies. However, we understand that the previously used terminologies and the imprecision in the text of some key sections have led the reader to an incorrect understanding of the scope of the work. Upon reviewing your comments, as well as those of the other reviewer, we have recognized this flaw in our manuscript presentation and have made a significant effort to thoroughly analyze the terminologies used and provide the necessary contextualizations, adjusting these notions throughout the text.

Referring to old Line 111: Participants for this study were recruited through verbal invitations, conducted by the researchers within the university environment, as well as through announcements in university newspapers, social media, and flyers posted within university premises. Exact demographics indicate that 76% of participants are undergraduate students, 20% are graduate students, and 4% are professionals with some affiliation with the university or friends of students. It is pertinent to emphasize that participation was entirely voluntary, and no participant received any form of financial compensation for their involvement in the study.

Referring to old Line 140: Previously, this information was included in lines 194-196. However, it has been relocated to a more appropriate subsection: "2.2. Materials and Equipment". Below are the relevant excerpts:

         Line 169: "We justify the choice of these emotions as the majority of research on expression recognition, from a categorical perspective, uses at least these four basic expressions with the additional neutral one".

Jack, R.E.; Garrod, O.G.B.; Schyns, P.G. Dynamic Facial Expressions of Emotion Transmit an Evolving Hierarchy of Signals over Time. Current Biology 2014, 24, 187–192. https://doi.org/10 823.1016/j.cub.2013.11.064.

         Line 250: "This number of images was aimed to mitigate visual fatigue facing the longest experiment [42]"

Abdulin, E.; Komogortsev, O. User Eye Fatigue Detection via Eye Movement Behavior. In Proceedings of the Proceedings of the 33rd Annual ACM Conference Extended Abstracts on Human Factors in Computing Systems; Association for Computing Machinery: Seoul, Republic of Korea, 2015; CHI EA ’15, pp. 1265–1270. https://doi.org/10.1145/2702613.2732812.

Referring to old Line 144: The selection of the image bank was based on the faces closely resembling our target audience in terms of age, around 30 years old, and balanced by gender, including 15 female and 15 male faces. We have incorporated the following information into the manuscript:

         Lines 167-178: "Thirty emotional facial images were presented as visual stimuli, equally distributed among the five emotional classes (depicting happiness, sadness, anger, fear, and neutrality). We justify the choice of these emotions as the majority of research on expression recognition, from a categorical perspective, uses at least these four basic expressions with the additional neutral one [4,25,35,36]. All images are frontal faces and were also equally distributed among genders (male and female), resulting in 15 images for each gender, with 3 facial images per gender-emotion. These images were sourced from The Karolinska Directed Emotional Faces - KDEF image database [37,38].  <…>  The selected facial images were oriented frontally, improving diagnostic region inspection. The individuals in the image database were Caucasian, approximately 30 years old, similar to the participants in this study."

Referring to old Line 181: We have included a detailed description in the "Materials and Methods" section outlining how the eye tracking induced visual field restriction with the moving window technique.

         In section "2.3. Procedure", lines 187-193: "The visual restriction was implemented using the moving window technique, directly controlled by the eye tracker in response to the viewer’s eye gaze. A mask was applied over the stimulus, with a visual aperture diameter corresponding to the projection of the respective visual angle on the screen, 5â—¦ for PFFV or 2â—¦ for FV. The diameter of the aperture for each condition is calculated beforehand based on the viewer’s distance and the monitor specifications (screen size and resolution), implying that the PFFV condition exhibits a larger aperture hole compared to the FV condition."

Referring to old Lines 186-187: We appreciate the observation. Indeed, our sentence was poorly formulated. We have rephrased it, resulting in the following explanation in the manuscript:

         Lines 232-228: "The choice of a free presentation time was justified by the scarcity of studies comparing emotional face judgment under different visual field restriction conditions. Although previous research has shown that facial recognition can be achieved with few fixations [41], our approach enabled the acquisition of a comparative dimension regarding the impact of participant judgment, facilitating a more comprehensive and meaningful analysis of participant responses across various experimental contexts."

Referring to old Line 214: The use of the term "weighted" was a mistake; the mean is not weighted. We have removed this term.

         When considering the mean and standard deviation alone, they are not ideal measures for Likert scale data. Likert scale data are ordinal, representing ordered categories rather than continuous intervals. Even when providing a symmetrical numerical scale with equal intervals between categories, calculating means and standard deviations can be misleading. This is because the distribution of Likert scale data often violates the assumptions of normality required for these measures.

         Regarding the advantage of considering Cns (Consensus) and Entropy over just a mean, Cns provides a measure of consensus or agreement among participants, while Entropy measures the uncertainty or diversity in responses. These measures offer a more nuanced understanding of the data compared to just looking at the mean, as they capture both the level of agreement and the variability in responses.

         Additionally, the overlap between the Cns and Mean curves in the previous Figure 2 is due to a misleading representation (as you reported in Question 3). While the Mean represents an indicator for the resultant concentration of answers, the Cns indicates the agreement between participants. These are distinct concepts. For example, if we have only 4 observations, two "Agree" and two "Disagree", the Mean is 0, indicating a resultant "Not sure" (according to Equation 3), while the Cns will give 0.79 (according to Equation 5), where Cns ranges from 0, total dissensus, to 1, total consensus. Although no one answered "Not sure", it acts as a qualitative resultant but gives no indication about the level of agreement, which is numerically given by the Cns.

         The entropy reflects the degree of disorder in the distribution of values, providing complementary insight into the variability and diversity within the dataset. However, it's important to note that unlike other measures, such as the consensus measure (Cns), entropy does not adhere to a closed numerical interval, thus requiring careful interpretation concerning the specific context of the data being analyzed.

         Given this context, we have adjusted the various sections of the manuscript to make these concepts clear to the readers.

         For more information on the Cns measure:

         Tastle, W.J.; Wierman, M.J. Consensus and dissention: A measure of ordinal dispersion. International Journal of Approximate Reasoning 2007, 45, 531–545. https://doi.org/10.1016/j.ijar.2006.06.024.

Referring to old Line 245: Following the example provided above, the mean represents a numerical value indicating a resultant concentration of answers. To interpret this value, it is necessary to establish appropriate numerical intervals corresponding to each possible label on the Likert scale. In the manuscript, these intervals are defined in Equation 3 (page 7). We refer to each label associated with one of these intervals as a "semantic counterpart", as it corresponds to one of the semantic values: "Strongly agree" (SA), "Agree" (A), "Not sure" (N), "Disagree" (D), "Strongly disagree" (SD).

Referring to old Line 264: We have revised the sentence to enhance clarity.

         Lines 331-338: "The EyeMMV algorithm is a dispersion-based method that employs spatiotemporal criteria, relying on two key parameters: tolerance (a spatial parameter) and minimum duration threshold (a temporal parameter). We set the tolerance to 40px, which corresponds to the projection of a circular area representing one degree of visual angle. This calculation takes into account the participant’s distance from the screen. Additionally, we established the minimum duration threshold at 80ms. These parameter choices are consistent with recommendations found in the literature for eye tracking methodology and eye movement analysis [16]."

Referring to old Line 266: We have revised the sentence to clarify the purpose of this parameter. Additionally, we have corrected a typo; it should be 80 ms instead of 100 ms.

         Lines 331-338: "The EyeMMV algorithm is a dispersion-based method that employs spatiotemporal criteria, relying on two key parameters: tolerance (a spatial parameter) and minimum duration threshold (a temporal parameter). We set the tolerance to 40px, which corresponds to the projection of a circular area representing one degree of visual angle. This calculation takes into account the participant’s distance from the screen. Additionally, we established the minimum duration threshold at 80ms. These parameter choices are consistent with recommendations found in the literature for eye tracking methodology and eye movement analysis [16]."

Referring to old Line 271-275: We originally intended to indicate that the metric mentioned could also serve the purpose of indicating areas of the face, as seen in the literature. However, in our experimental design, we did not use it for this purpose due to the specific use of the moving window paradigm and different visual field restriction conditions. Therefore, we have rephrased this passage to avoid ambiguity and maintain clarity regarding our scope.

Referring to old Line 281: We have revised this passage to provide a clearer and more concise explanation.

         Lines 343-345: "the average fixation duration is calculated as the sum of fixation durations divided by the number of stimulus presentations. Providing an indication of cognitive function activity, with longer durations potentially reflecting heightened cognitive engagement [48]."

Referring to old Line 284: The statistical analysis encompassed various metrics, including recognition accuracy, visual inspection time, number of fixations, and fixation duration as dependent variables, with NVR, PFV, and FV as primary independent variables. In emotional judgment analyses, the same metrics are used as dependent variables, with expression types and NVR, PFV, and FV as independent variables, where expression types serve as secondary independent variables.

Referring to old Lines 290, 298: In the given context, the dependent variables refer to all the dependent metrics considered in the subsequent statistical analyses (recognition accuracy, visual inspection time, number of fixations, fixation duration).

This specific passage establishes the appropriate statistical framework for the analyses developed later in the results section, including the tests used, significance value, and post-hoc testing, clarifying the comparison methods adopted and the significance level for rejecting the null hypothesis. Observe that the passage has been slightly adjusted and moved to the beginning of the "Data Analysis and Measures" subsection within the "Materials and Methods" section for better organization and clarity.

Q3 RESPONSE: We have thoroughly reviewed the entire results section, making appropriate adjustments to enhance clarity and maintain consistency in the terminology used.

Referring to old Lines 317-318: We have revised the passage for clarity.

         Lines 410-413: "In the NVR experimental condition, lower Cns values of 0.77 and 0.78 were observed for fear and neutrality, respectively, with corresponding entropy values of 0.95 and 1.02. In the PFFV and FV experimental conditions, the lowest Cns values were 0.62 and 0.49, respectively, both for neutrality, with corresponding entropy values of 1.09 and 1.00."

Referring to old Lines 326-328: We adjusted the paragraph for greater clarity and conciseness. The concepts utilized in this analysis are described in the "Materials and Methods" section, on page 7.

         Lines 427-428: "It is evident that all expressions, in all conditions, had a predominantly high semantic concentration in "Strongly Agree" in their choices."

Figure 2 Consideration: You are correct; a line graph is a misleading representation for this type of data. Further details regarding this have been included in response to Question 2, under "Referring to old Line 214".

         Additionally, we have adjusted the figure by including standard deviation marks on the mean measurements and have removed the tables associated with the figure.

Other Considerations: We have added notations of statistical significance to various figures in the manuscript. We have also checked all the graphs to include units where they were not present, except in those with purely numerical scales.

Q4 RESPONSE: We standardized the nomenclature for facial expressions throughout the manuscript.

Old line 5: We standardized the nomenclature of the experimental condition to NVR when referring to "no visual restriction".

Old line 9: We included an explanation for the abbreviation MMV (Eye Movements Metrics and Visualizations) and its application in lines 327-338.

Old line 9: "Preliminary results" was a typo and has been removed.

Old line 69: The same situation occurred with "will", a typo that has also been removed.

Old lines 192-193: We adjusted the terminology in the text, in the "Procedure" subsection, to correctly refer to what is presented in Figure 1; lines 243-245. And we also informed in the "Data Analysis and Measures" subsection the nomenclature adopted for the analysis of this data, to standardize it with that used in other studies; lines 278-284.

Old line 202: The sentence was wordy. We reformulated it.

Old lines 269-270: We reformulated this entire paragraph.

Old line 385: This expression was wordy. We removed it.

Old line 399: We adjusted the sentence.

Old line 433: It refers to the modulation of the visual angle. We adjusted the sentence to clarify.

Old line 437: We adjusted the sentence.

Round 2

Reviewer 1 Report

Comments and Suggestions for Authors

The authors have made more specific revisions to the manuscript, and the content of the manuscript has been clarified. It is suggested to add discussion of the limitations of the study.

(1) The study set PFFV within 5 degrees, why not examine 3 degrees or 4 degrees? Future studies can set multiple levels of PFFV to examine the differences between them, and at what degree does it reach saturation (that is, the accuracy no longer increases)?

(2) The study obtained contributions from PFFV, which may include contributions from parafoveal vision (PFV), FV, and the interaction between PFV and FV. In the future, if possible, the condition of parafoveal vision (2-5 degrees) should be set to isolate the contribution of PFV and the interaction between PFV and FV.

Author Response

In response to your comments, we have included two new paragraphs discussing the limitations of our study in the revised manuscript. We are confident that addressing these limitations will contribute to a more comprehensive understanding of the effects of visual field restriction on facial expression recognition and clarify future perspectives on this research problem.

Below, we provide the copy of these paragraphs.

"Nevertheless, the limitations of this study mainly arise from implications for the determined criteria to restrict the visual field for parafoveal vision. Firstly, the choice to examine the PFFV with a 5° aperture limits our understanding of specific characteristics of visual processing within the parafoveal extension (2° to 5°). The fixed 5° PFFV aperture hinders the evaluation of a more precise point of saturation for observed differences in accuracy between experimental conditions. Secondly, it's important to acknowledge that the observed contributions of PFFV in this study include influences from parafoveal and foveal information simultaneously, thus limiting the understanding of purely parafoveal vision contributions. For example, it is not possible to ascertain whether the observable changes were mostly due to the parafoveal contribution or occurred only when the fovea and parafovea were simultaneously engaged.

Future research could address the first limitation by exploring intermediate levels of PFFV restriction to assess their impact on accuracy and to identify the saturation point more precisely, at which accuracy stabilizes and no longer increases. This would allow for a more comprehensive understanding of the effects of PFFV interplay according to the increase in visual field aperture and its influence on perceptual processes. Regarding the second limitation, isolating the contributions of parafoveal vision could be beneficial to assess the interaction dynamics between parafovea and fovea more clearly. This could be achieved by setting specific conditions for isolated parafoveal vision (e.g., central occlusion with 2° to 5° aperture) to independently evaluate its role and interaction with foveal vision. In doing so, we could enhance our understanding of how these different visual components contribute to visual perception and their interplay mechanisms."

Thank you once again for your valuable input and for your attention to our manuscript.

Reviewer 2 Report

Comments and Suggestions for Authors

1)      First of all, I must point out that the Authors’ reply letter does not accurately list all the changes they made (i.e., they should quote the precise sentences), but mostly reports only general statements (e.g.: we changed, we rephrased, etc). From a reviewer point of view, this makes the re-review difficult and time consuming.

2)      Some of the explanations in the reply letter are not reported in the manuscript; see, e.g, the explanation referred to “old line 111” (in the new manuscript, line 138): all the explanations should be added to the manuscript, as the information is not to be intended as just a reviewer’s request, but as useful data for every reader.

3)      Line 169 (old line 140): the explanation does not hold, as at least one of the quoted papers (36) mentions all six classical emotions, while another one (25) claims that only four basic emotions are identifiable, which is an entirely different concept and should be mentioned in the introduction a theory alternative to Ekman’s.

4)      About “Old Lines 271-275” (new 345-349): the new sentence “This measure offers insights into the quantification of visual attention across facial characteristics during visual exploration” is as misleading as the old one, in that it is not clear how number of fixations could possibly provide insights into attention towards different facial characteristics. The Authors should either clarify this statement, or delete it.

5)      Referring to old Line 284: The reply letter and the revised manuscript report different statements, specifically, in the revised manuscript facial expressions are still listed as dependent variables (Lines 264-265: “The statistical analysis encompassed various metrics, including number of fixations, fixation duration, visual inspection time, and recognition accuracy, with expression types as dependent variables...”). Not only this is inconsistent, but it appears downright wrong; the Authors should explain or, more likely, correct.

6)      Referring to old lines 290, 298 (now, see new line 272): if the Authors make general statements about all dependent variables, they should use the plural “dependent variables”, or the expression “each dependent variable”, or similar; if not, they should identify the specific variable they refer to

7)      Old Line 433 (new 682): In spite of what the Authors claim in their letter, the sentence has not been changed and is as obscure as it was in the previous version.    

Other points:

Line 339: please explicate px (in “40px”)

Line 403: dissensus: dissensus??

Comments on the Quality of English Language

Some mistakes or awkward sentences, please have the text revised by a native speaker

Author Response

We are again grateful to the editor and the referee for their insightful comments and a new opportunity to improve our manuscript quality.  In the revised manuscript, we have addressed all the comments and suggestions for improving our paper. The changes and responses are detailed, as follows.

RESPONSE Q1: We sincerely apologize for any inconvenience caused by the lack of specificity in our previous response regarding the changes made to the manuscript. During the first review phase, we made extensive revisions to the manuscript based on your valuable and detailed feedback. In an attempt to keep our reply letter concise, some sentences were not fully quoted, leading to the oversight you've pointed out.

         We understand the importance of providing clear and specific details about the modifications made to the manuscript, and we will ensure that our responses in the future include precise quotations of the sentences or sections that were modified or rephrased.

         Once again, we apologize for any oversight on our part and appreciate your patience and understanding as we work to address your concerns.

            Please note that we are using LatexDiff to automatically highlight the changes in the manuscript, with additions marked in blue and removals marked in red.

RESPONSE Q2: We have included the proper explanation in the manuscript.

         Lines 158-164: “Participants were recruited through verbal invitations extended by the researchers within the university environment, as well as through announcements in university newspapers, social media, and flyers posted on university premises. Demographic data indicates that 76% of the participants were undergraduate students, 20% were graduate students, and 4% were professionals with some affiliation with the university or friends of students. It is pertinent to emphasize that participation was entirely voluntary, and no participant received any form of financial compensation for their involvement in the study.

RESPONSE Q3: We have revised the section in question to provide a clearer explanation. We have decided to retain the reference to paper [36] but have adjusted its placement in the text to better align with its context. Paper [36] is a review article that discusses studies involving all six classical emotions as well as the proposition of the four basic emotions [25] as an alternative to reduce recognition confusion, providing important background information for our study.

         Lines 176-192: “Thirty emotional facial images were presented as visual stimuli, equally distributed among the five emotional classes: happiness, sadness, anger, fear, and neutrality. We justified the selection of these emotions based on research findings by Jack et al. [25], who identified evidence for four well-discriminated basic expressions. Their study, utilizing a method combining perceptual expectation modeling, information theory, and Bayesian classifiers, suggested that fear and surprise share processing and representation codes, as do disgust and anger, particularly at early processing stages, aligning with the reported patterns of major recognition confusions in categorization tasks [36]. Furthermore, these four emotions are encompassed within Ekman's six classical emotions [3,37]. In our study, the alternative theory of four basic expressions reduces confusion in emotion recognition by prioritizing more identifiable emotions [25,36,38], which is crucial for enhancing effects stemming from changes in visual field restriction for the recognition of facial expressions. Finally, the inclusion of the neutral face is a common practice in facial expression recognition studies, serving as a neutral and stable reference for comparison with emotional expressions [36].

RESPONSE Q4: We opted to delete it.

RESPONSE Q5: Thank you for bringing this to our attention. We appreciate your meticulous review. Indeed, there was an error in our textual description regarding the facial expressions as dependent variables. We have corrected this text excerpt to accurately reflect the role of facial expressions in our analysis. The revised statement is as follows:

         Lines 281-285: “The statistical analysis encompassed various metrics, including the number of fixations, fixation duration, visual inspection time, and recognition accuracy, treated as dependent variables, while NVR, PFFV, and FV were considered independent variables. Facial expressions were categorized as factors influencing these dependent variables, allowing for statistical data analysis segmented by facial expression.

RESPONSE Q6: We acknowledge that it was indeed a typographical error. The word has been corrected to accurately reflect the intended meaning: “dependent variables”. Thank you for bringing this to our attention.

RESPONSE Q7: We have revisited the sentence and made the necessary revisions to improve its clarity. Thank you for bringing this to our attention, and we appreciate your diligence in ensuring the clarity of our manuscript.

         In regard to what is being modulated in the study comprising images of faces and houses: Modulation of discrimination performance of stimuli according to eccentricity.

         Lines 694-703: “For instance, Kreichman et al. [63] demonstrated that the impact of eccentricity (how far away an stimulus is from the point of fixation) on the recognition of face images differs from its impact on house images. Specifically, their study found that faces are more affected by eccentricity, leading to reduced accuracy in face discrimination tasks compared to house discrimination tasks at greater eccentricities. This finding suggests that the ability to perceive and recognize faces declines more rapidly with increasing eccentricity. However, it is important to note that the methodologies employed in their study differ from ours.

         Note that we found an error in the citation. The study by Fiorentini is related to the sentences following this text excerpt. We have made the necessary adjustments by citing Kreichman in the text excerpt mentioned in the question and citing Fiorentini in the subsequent text excerpt (now in a new paragraph).

RESPONSE Other Points:

         Old Line 339 (now, line 357): “We set the tolerance to 40 pixels, which corresponds to the projection of a circular area representing one degree of visual angle on the screen.

         Old Line 403 (now, line 442): The word “dissensus” was not necessary, so it was removed to focus on a concise text.